# Effects of Missing Data on Heart Rate Variability Metrics

**DOI:** 10.3390/s22155774

**Published:** 2022-08-02

**Authors:** Diego Cajal, David Hernando, Jesús Lázaro, Pablo Laguna, Eduardo Gil, Raquel Bailón

**Affiliations:** 1Biomedical Signal Interpretation and Computational Simulation (BSICoS) Group, Aragón Institute of Engineering Research (I3A), IIS Aragón, University of Zaragoza, 50018 Zaragoza, Spain; dhernand@unizar.es (D.H.); jlazarop@unizar.es (J.L.); laguna@unizar.es (P.L.); edugilh@unizar.es (E.G.); rbailon@unizar.es (R.B.); 2Centro de Investigación Biomédica en Red en Bioingeniería, Biomateriales y Nanomedicina (CIBER-BBN), 28029 Madrid, Spain

**Keywords:** HRV, ANS, Apple Watch, Poincaré plots

## Abstract

Heart rate variability (HRV) has been studied for decades in clinical environments. Currently, the exponential growth of wearable devices in health monitoring is leading to new challenges that need to be solved. These devices have relatively poor signal quality and are affected by numerous motion artifacts, with data loss being the main stumbling block for their use in HRV analysis. In the present paper, it is shown how data loss affects HRV metrics in the time domain and frequency domain and Poincaré plots. A gap-filling method is proposed and compared to other existing approaches to alleviate these effects, both with simulated (16 subjects) and real (20 subjects) missing data. Two different data loss scenarios have been simulated: (i) scattered missing beats, related to a low signal to noise ratio; and (ii) bursts of missing beats, with the most common due to motion artifacts. In addition, a real database of photoplethysmography-derived pulse detection series provided by Apple Watch during a protocol including relax and stress stages is analyzed. The best correction method and maximum acceptable missing beats are given. Results suggest that correction without gap filling is the best option for the standard deviation of the normal-to-normal intervals (SDNN), root mean square of successive differences (RMSSD) and Poincaré plot metrics in datasets with bursts of missing beats predominance (p<0.05), whereas they benefit from gap-filling approaches in the case of scattered missing beats (p<0.05). Gap-filling approaches are also the best for frequency-domain metrics (p<0.05). The findings of this work are useful for the design of robust HRV applications depending on missing data tolerance and the desired HRV metrics.

## 1. Introduction

For several decades, heart rate variability (HRV) has been a researched field because of its ability to evaluate the autonomic nervous system (ANS) noninvasively, presenting itself as a potential tool for the prognosis, diagnosis and monitoring of diseases, mainly in the clinical environment [1,2,3,4,5,6,7]. HRV is defined as the changes in the duration of the beat-to-beat interval, which is calculated from R-wave detections in electrocardiographic (ECG) signals. Alternatively, variability in pulse rate (PRV) can be derived from pulse photoplethysmography (PPG). This signal can be recorded at various locations on the body, making it of interest for wearable devices. Despite pulse rate variability being different from HRV, it can be used as a surrogate in many practical situations [8,9].

The exponential growth of wearable devices able to record ECG and/or PPG signals has opened up a new horizon for HRV, allowing massive monitoring at a relatively low cost. The accessibility of a large variety of designs has made them an everyday use tool, allowing non-invasive health monitoring in the general population. In this context, assessing the state of the ANS during daily life has become a very attractive objective in the field of health and well-being. However, obtaining reliable variability measurements from wearable devices is challenging. Wearable devices are worn throughout the day in constantly changing conditions, and motion artifacts are very frequent. In addition, comfortability is relevant when deciding the place of recording of a wearable device, in contrast to the clinical settings, where the signal quality is usually more relevant. All this leads to an overall low signal quality compared to clinical monitoring scenarios, downgrading the performance of the traditional HRV methods. Most devices only measure the mean heart rate (MHR), which is very robust to data loss in stationary conditions but less powerful for ANS assessment than HRV. Although changes in the MHR are mainly induced by the ANS, it cannot be considered a measure of autonomic function [10,11,12]. Despite studies that criticize the added value of HRV with respect to MHR [13], there are scenarios in which an alteration of ANS function produces changes in HRV but not in MHR, such as in depressed patients with respect to controls [14] or in exercise contexts [15].

Acquisition technology has made a qualitative leap that has surpassed traditional HRV preprocessing methods to some extent. In a few years, the challenge has shifted from dealing with casual artifacts to being forced to forego a large part of the total recording time. The proliferation of health applications of wearable devices makes it necessary to investigate the degradation of HRV metrics in the presence of incomplete recordings, as well as new methods that allow robust analysis under adverse conditions.

### 1.1. Related Work

Artifacts have been a concern since the beginning of HRV studies, as they can appear even in the most controlled environments. Most of the works in the literature focus on artifacts of small duration, which are often treated in the same way as ectopic beats [16,17,18,19,20,21,22]. In general, methods are divided between those that simply remove outliers in beat detections—both false positives and false negatives—and those that interpolate them based on accepted proximal values (gap-filling methods) [16]. Correction methods are mandatory since errors representing less than 0.1% of the detections may cause variations of up to 50% in some HRV metrics [16].

Some gap-filling methods generate evenly-spaced interpolations. The beat event series is not available with these methods, so time-domain metrics or Poincaré plots cannot be assessed. Mateo and Laguna proposed an IPFM-model based corrector for ectopic beats on the heart timing signal [17], a continuous signal, assuming that autonomic modulation can be modeled using a band-limited signal. Meanwhile, McNames et al. used an impulse rejection filter on the instantaneous heart rate signal—evenly sampled—on the basis that nonpathological artifacts are of small duration and large amplitude [18]. Lee and Yu detected and corrected outliers in the tachogram using cubic splines [19].

On the other hand, some studies obtain a corrected unevenly-sampled inter-beat interval (IBI) series, allowing the assessment of time-domain metrics and Poincaré plots. Begum et al. used k-nearest neighbors in the IBI series [20], while Al Osman et al. used a combination of cubic and nonlinear predictive interpolation methods [21]. An interesting aspect of the latter is the use of simulation to introduce artifacts in order to compare errors. Giles and Draper compared different interpolation methods of the IBI series, including cubic splines [22].

Although the previous methods may work for isolated outliers, they have not been evaluated for longer artifact segments. Baek and Shin studied the degradation of temporal and frequency metrics in response to an increase in missing IBI data, obtained by simulation, although they do not provide any correction method [23]. The simulation randomly removes samples from the tachogram in an increasing manner, over a fairly wide range, from 5 to 285 intervals, in 5 min recordings. Morelli et al. developed one of the first studies to investigate the effect of large heartbeat losses, from the perspective of wearable devices [24]. Their simulation method for missing detections is based on a two-state Markov chain, simulating losses of 30%, 50% and 70% of IBIs. This is one of the most complete studies on artifact correction applied to wearables, including temporal and frequency metrics and Poincaré plots. Benchekroun et al. used filtering and gap filling using a Gaussian distribution in IBI series with 5% to 35% simulated missing beats [25]. HRV metrics were derived from corrected series and used as features for a stress/relax classification. Classification results were compared with other gap-filling approaches (linear, spline, and pchip). Nevertheless, no separate metric results were reported. Królak et al. proposed a gap-filling algorithm tested with bursts of up to seven missing beats [26]. They reported that cubic interpolation can in some cases result in lower errors for long gaps. Finally, some works address artifact correction in the detection stage, using methods such as adaptive filtering, wavelet transform or feature extraction of the cardiac signal [27,28]. These approaches are beyond the scope of this paper, as they are signal specific, and many wearables do not allow exporting cardiac signals but event series. In addition, they can be used in conjunction with event series correction.

### 1.2. Aims of the Study

There is still much to be known about the degradation of HRV metrics in scenarios with large missing data. To the authors’ knowledge, there is no study that provides insight into how correction methods behave under different types of losses that can occur in a real case: bursts and scattered missing beats. There is also no conclusion on the maximum burst size to discard a segment for further analysis. The same is true for scattered missing beats. In this work, the degradation of different HRV metrics—in the time domain and frequency domain and Poincaré plots—is evaluated in missing data scenarios. A missing data simulation protocol has been developed for this purpose. In addition, a method to attenuate the effect of missing data in HRV metrics has been proposed and compared to existing methods in the literature. Then, these methods have been applied to analyze PRV derived from Apple Watch. This work aims to contribute to HRV/PRV analysis by proposing guidelines to select the best correction method for each studied metric and missing data scenario and to provide conclusions about when to discard a segment for further analysis depending on the quantity and distribution of missing data.

## 2. Materials and Methods

### 2.1. Simulation of Missing Beats

The simulation study was based on a real database comprising 16 subjects (age 28.5±2.8 years, 10 males) who underwent a tilt-table test consisting of the following: 4 min in supine position, 5 min at a 70º angle and 4 min back to supine position. An ECG signal—V4 lead—was recorded using Biopac’s ECG100C amplifier and disposable Ag–AgCl electrodes with a sampling frequency of 1000 Hz. See [8] for further details. Two 2 min duration segments, free of artifacts and ectopic beats, were selected for each subject: one for the first supine stage and the other for the tilt stage. Stationarity was assumed for this duration [8]. HRV metric degradation was evaluated in terms of error, as well as in their ability to distinguish the tilt and supine states, characterized by changed sympathovagal balance.

A wavelet-based algorithm was used for QRS detection [29]. Detections were visually inspected and corrected if necessary. First, HRV metrics were computed prior to data removal, resulting in a benchmark for each method under review. Then, missing beats were simulated by removing detections from the time series in two ways: (1) by a random selection using a binomial distribution and (2) through deletion bursts, with an increasing number of missing beats in each one. The former simulated the effect of a low signal-to-noise ratio (SNR). Sometimes, signals had sufficient quality to perform detections, although an automatic detector could still miss some pulses in borderline situations. A binomial distribution was used, so every beat was deleted with a *p* probability, i.e., every beat deletion was an independent Bernoulli trial. Ten different realizations of this stochastic process were computed for each segment, obtaining a total of 160 segments for each supine and tilt position. Figure 1a shows an example of a 40-beat segment, in which 25% of the samples are removed (p=0.25). In successive realizations, the positions of the removed samples changed randomly. On the other hand, artifacts could affect signals even with a high long-term SNR. Movements were mainly the cause of this kind of noise—a common problem in wearables—characterized by a finite duration and a total masking of the physiological signal. These events caused a burst of missing detections. This effect was simulated by removing central elements from the series with windows of a certain duration. Although it is possible to find bursts at any position by taking random segments of a signal, bursts were not simulated at interval ends, since the most advisable solution in that case is not to use those first or last seconds of the window. Specifically, 30 s at each of the two segment ends was not considered for removal. Beat removal was restricted to the the remaining segment. Samples were removed from segments with a sliding window of 10 steps, again obtaining 160 segments per supine/tilt position. An example is shown in Figure 1b. As the duration of the bursts was determined in seconds, different numbers of beats were removed at each step depending on the instantaneous heart rate, even for the same segment. For simplicity, in the figure, all bursts have the same number of elements. In scenarios with scattered missed beats, an increase in missed beats poses a challenge in detecting where each missed beat is located, as the baseline can be lost. However, if a correct detection has been made, correction is still straightforward as adjacent beats are present. On the other hand, in the case of bursts of missing beats, detection becomes easier the greater the number of missing beats, as they will produce a larger outlier. In this case, the complication lies in finding out how many beats are missing and how to perform corrections based on gap-filling methods.

Scenarios with possible extra detections (false positives) are not analyzed in this work. This decision was made on the basis that only a few false positives would complicate any correction due to loss of reference. Therefore, it is assumed that detections should be performed after a signal quality evaluation stage that is sufficiently restrictive to avoid most false positives.

### 2.2. Apple Watch Dataset

As a real case, the dataset described in Hernando et al. in [30] was selected. It is composed of 20 healthy subjects (age 31.3±8.2 years, 12 males) who underwent a protocol that involved controlled relax and stress environments. Three two-minute-length segments per subject were used—the same duration as the simulation—for each relax and stress phase, yielding a total of 120 segments. Two heart rate-related series were obtained in each segment: PPG-based pulse detection series recorded by the Apple Watch on the wrist, and ECG-based R-wave detection series recorded by Polar H7 (Polar Electro Ltd., Kempele, Finland), with the last used as benchmark. It is worthwhile to note that Apple Watch outputs the event timestamps only when the internal PPG allows reliable pulse detection according to an internal signal quality algorithm. Thus, the derived pulse-to-pulse series present intermittent gaps. A total of 206 gaps were found in the recordings, equivalent to 1321 missing intervals. Missing data represent around 10% of total events, distributed in gaps of 6 s length on average. The minimum gap length is 3.3 s, and the maximum is 10.4 s. Synchronization between Apple Watch and Polar H7 was performed using a delay that maximized the cross correlation using the first 20 intervals, where no gaps appeared in Apple Watch recordings [30].

### 2.3. Missing Data Detection

Figure 2 displays a graphical summary of the methods applied, described in Section 2.3, Section 2.4, Section 2.5. Missing data detection is usually based on detecting physiologically abnormal increases in the interval series that suggest that at least one heartbeat is missing. In this study, interval series are represented using the interval function $d_{IF}\left(t_{k}\right)$, defined by
(1)$$d_{IF}\left(t_{k}\right)=\sum_{k=1}^{K}\left(t_{k}-t_{k-1}\right)\delta \left(t-t_{k}\right)$$
where $t_{k}$ is the event series. This function is defined on a continuous-time basis, with zero values for all *t* other than $t_{k}$; for example, each event occurring at time $t_{k}$ is represented by a unit impulse function $\delta (t-t_{k})$ scaled by the length of the preceding interval [31,32]. The scaling causes missing beats to produce outliers in $d_{IF}\left(t_{k}\right)$ at each $t_{k}$ corresponding to events after a gap. A moving median threshold is used as outlier detection (OD) rule. First, $d_{IF}\left(t_{k}\right)$ is filtered with a 2Lth-order median filter to produce an expected inter-beat interval (EIBI) value for each event $t_{k}$ [21]:(2)$$EIBI\left(t_{k}\right)=median\left(\left\{d_{IF}\left(t_{i}\right)\;|\;i\in N,\;\left(k-L\right)<i\leq \left(k+L\right)\right\}\right)$$

The interval at $t_{k}$ is marked as an outlier if the equation
(3)$$d_{IF}\left(t_{k}\right)>\left(\alpha \times EIBI\left(t_{k}\right)\right)$$
is satisfied, i.e., if the the interval is longer than α times the expected interval, with α∈[1,∞). The values of α and *L* were empirically set using the simulation dataset, resulting in α=1.5 and L=25. The best value for α was searched between 1 and 1.7 with a step of 0.1. Similarly, the best value for *L* was searched between 5 and 50 with a step of 5.

### 2.4. Correction Methods

The simplest correction rule is to remove outliers from $d_{IF}\left(t_{k}\right)$. This method is referred to as Outlier Removal (OR) in this paper, and its estimations are denoted $t_{k}^{\mathrm{OR}}$. However, some metrics are greatly affected by incomplete interval series. Thus, methods for estimating missing beat locations remain very interesting. A novel gap-filling method is proposed as follows. First, missing beats are estimated by interpolation allowing a single beat per gap. The outlier detection rule (Equation (Equation 3), Section 2.3) is applied to each new estimate, setting α=1.1 for a better fit. If a gap is still detected, the algorithm discards the added beats and passes to the next gap. In the next iteration, it will try to fill it with one more beat. Otherwise, it is checked if $d_{IF}\left(t_{k}\right)>\left(\beta \times EIBI\left(t_{k}\right)\right)$ for all the added $t_{k}$, with β=0.9, to avoid introducing more beats than necessary. If this condition is not fulfilled, the gap is filled with the number of beats from the previous iteration and marked as corrected. Both α and β were empirically set using the simulation dataset. The best value for α was searched between 1 and 1.5 with a step of 0.1, while the best value for β was searched between 0.5 and 1 with a step of 1. At the end of the iteration, i.e., when all the gaps were covered, the outlier detection rule was checked again in the whole segment. If it did not pass, a new iteration was started, using one more beat per gap until the segment was completed. A flowchart of this algorithm is presented in Figure 3.

The interpolation method will greatly affect the results. Here, both linear interpolation and non-linear interpolation by Hermite polynomials were used. Hermite polynomials preserve data shape and have already been shown to outperform other methods in HRV gap-filling applications [33]. Hereafter, gap-filling methods are referred to as linear (L) and non-linear (NL) gap filling and their estimations $t_{k}^{L}$ and $t_{k}^{\mathrm{NL}}$, respectively.

Finally, the correction method described by Mateo and Laguna in [17] has also been used when analyzing metrics in the frequency domain using Fourier-based techniques. This method is referred to as model-based (M) correction. OR, L, NL and M corrections are used both in scattered missing beats and bursts.

### 2.5. HRV Metrics

Metrics in the time, frequency and Poincaré-related domain have been computed.

*Time domain*: Mean heart rate (MHR), standard deviation of the normal-to-normal interval (SDNN) and root mean square of successive differences (RMSSD), as described in [1].*Frequency domain*: LF and HF powers ($P_{LF}$,$P_{HF}$); LF power measured in normalized units ($P_{LFn}$); and $P_{LF}$/$P_{HF}$ ratio. Only relative errors of $P_{LF}$ and $P_{HF}$ are presented, as the other two are derived from them. While all subjects are included when measuring relative errors, not all of them could be included when measuring the ability to distinguish sympathovagal balance. For this comparison, only subjects with respiratory rates above the classic LF band (>0.15 Hz) were selected, thus allowing a correct frequency component separation [34]. Therefore, simulation dataset is reduced from 16 to 9 subjects (age 28.3±2.6 years, 5 males). This selection only applies when comparing metrics in the frequency domain. No selection is made in the Apple Watch dataset. In addition, respiratory rate does not exceed 0.4 Hz—the classic HF band upper limit—in any case.Spectral estimation is performed via Fast Fourier Transform (FFT) and Lomb’s methods. FFT estimations are made on the evenly-sampled instantaneous heart rate signal, r(t), obtained from the IPFM model [17]. This model assumes the ANS modulates the sinoatrial node by a band-limited zero-mean signal [35]. In [36], it is shown that spectra derived from r(t) are a more accurate estimator for HRV than spectra derived from evenly-sampled interval series, avoiding spurious components and low-pass filtering effects. Welch’s method is used for periodogram averaging using 60 s Hamming windows with 50% overlap. For 120 s signals, three periodograms are averaged. Powers are computed using trapezoidal integration and classic windows (0.04–0.15 Hz for LF and 0.15–0.4 Hz for HF). This FFT-based approach has been tested using both model-based and gap-filling correction.On the other hand, Lomb’s periodograms can be computed from unevenly spaced signals, even in the presence of missing beats. Therefore, this method has been tested both using OR and gap-filling correction. It is demonstrated that the estimates on the heart rate representations are more accurate than on the beat interval representations [36]; therefore, Lomb’s periodograms are computed on the inverse interval function
(4)$$d_{IIF}\left(t_{k}\right)=\sum_{k=1}^{K}\frac{1}{\left(t_{k}-t_{k-1}\right)}\delta \left(t-t_{k}\right)$$
obtained by inverting the intervals of $d_{IF}\left(t_{k}\right)$ after correction. Lomb’s periodograms are averaged using 60 s Hamming windows with 50% overlap, and powers are computed using trapezoidal integration within the classic windows as well.*Poincaré plots*: SD1, SD2, SD1/SD2, ellipse area (S=π·SD1·SD2), mean distance to the ellipse centroid (Md) and standard deviation to the ellipse centroid (Sd) have been computed using the ellipse fitting method [37]. As S and SD1/SD2 are computed from SD1 and SD2, relative errors are not shown for these metrics. The reliability of Poincaré plots in ultra-short term segments—less than 5 min, as this case—has been demonstrated recently [38].

### 2.6. Statistical Analysis

Relative errors (ϵ) have been computed as the absolute value of the difference between the reference and the correction divided by the reference value, both in the simulation study and in the real database. Values are expressed as a percentage. In the simulation case, ϵ is obtained for each correction method, and within each method for each type and number of removed beats. In the Apple Watch case, only one ϵ is shown for each method, since missing beats are given by the dataset (Section 2.2). ϵ is presented as a tuple of three elements: *median (first quartile–third quartile)*. A Wilcoxon signed rank test has been performed to compare the performance of methods on the same segments.

On the other hand, another signed rank test has been applied for ANS state discrimination results. The test is done to supine/relax and tilt/stress records as separate samples, pairing states from the same subject. Metrics that could not differentiate states in any case have been omitted. Also coverage graphs are shown for the Apple Watch dataset. These graphs show the percentage of cases ($n_{th}$) with a relative error under a certain threshold ($\epsilon_{th}$) as $\epsilon_{th}$ is increased. These results can be very valuable to choose a correction method depending on the allowed tolerance of each application. Coverage graphs are not included for the simulation because of the large number of combinations depending on the type and number of deletions. Segment rejection decision thresholds, i.e., the maximum deletion probability/burst duration allowed to obtain reliable results, are also proposed in Section 4. These thresholds are proposed based on the criterion that the third quartile of the relative error does not exceed 20%.

## 3. Results

### 3.1. Time-Domain Metrics

Table 1 shows the relative error values of the different metrics with increasing deletion probability in the case of scattered missing beats (Table 1a) and burst duration (Table 1b). Regarding the relative error of scattered missing beats, NL gap filling is the best-performing correction method for MHR and SDNN for all deletion probabilities, although no significant differences can be found between OR and NL with up to 35% missing beats in the case of MHR. L gap filling yields the best results for RMSSD up to 25% deletion probability. A higher degradation can be observed at high loss rates, with OR the best option from 25% deletion probability onwards. In the case of bursts, NL gap filling yields the best results for MHR up to 10 s bursts. No significant differences can be found between OR and NL from 15 s. OR gives the best results for SDNN and RMSSD.

Table 2 shows the results of the Wilcoxon test for distinguishing between supine and tilt states. The first column shows the reference test results. There are no major differences between methods, although only NL is able to maintain the benchmark results throughout the entire simulation. This is consistent with the results of NL in terms of relative error. Both OR and L fail with SDNN in the case of scattered missing beats with deletion probabilities greater than 15%.

Table 3 shows the Apple Watch dataset’s relative errors, exhibiting equality among all correction methods. Figure 4 shows the coverage from the Apple Watch dataset. No differences are found between methods. MHR once again demonstrates great robustness, with an $n_{th}$ close to 100% with less than 2% $\epsilon_{th}$. SDNN achieves 80% $n_{th}$ with 10% $\epsilon_{th}$, while for the same $\epsilon_{th}$, RMSSD has 60% $n_{th}$. Finally, Figure 5 shows metric distributions with relax (green) and stress (blue) groups separately from the Apple Watch dataset. Wilcoxon test results are marked with asterisks above each pair. One asterisk indicates p<0.05, and two asterisks indicate p<0.001. All correction methods present the same behavior for MHR and SDNN. RMSSD results show improved OR performance by maintaining the reference p<0.001 versus p<0.05 of the gap-filling methods.

### 3.2. Frequency-Domain Metrics Computed via FFT

In the case of frequency-domain metrics, gap-filling methods show a clear improvement. NL gap filling is the best-performing method in terms of relative error in the case of scattered missing beats (Table 4). The correction advantage of gap-filling is maintained in the case of bursts. Although differences are reduced, they are still significant. In addition, differences between L and NL gap filling are reduced. In this case, L gap filling performs better for $P_{HF}$, while NL is still better for $P_{LF}$. Another aspect to note is that correction is not as effective in $P_{HF}$ as in $P_{LF}$ with scattered missing beats. Discrimination results follow a similar pattern (Table 5). For scattered missing beats, gap-filling correction performed better than M correction for $P_{HF}$ and $P_{LF}$/$P_{HF}$. This difference only appears after a 35% deletion probability; thus, the differences are not very large. On the other hand, results are identical for the burst case. $P_{LF}$ showed no discrimination capacity for this dataset.

Regarding the Apple Watch dataset, NL gap filling obtains the best performance at low frequencies (Table 6), although there is virtually no difference at high frequencies. In addition, $P_{HF}$ errors are higher than $P_{LF}$ errors as in the simulation. Coverage graphs show the same phenomena (Figure 6). $P_{LF}$ coverages are similar until 10% $\epsilon_{th}$—approximately 60% $n_{th}$—separating thereafter. NL gap filling is the best correction method, followed by L gap filling. In contrast, there are no differences for the $P_{HF}$ case. In addition, the coverage is clearly lower, approximately 40% $n_{th}$ at 10% $\epsilon_{th}$. Both $P_{LF}$ and $P_{HF}$ correctly discriminate the states (Figure 7), showing no difference between correction methods.

### 3.3. Frequency-Domain Metrics Computed via Lomb’s Method

In the case of frequency-domain results calculated via Lomb’s periodograms, NL gap filling clearly outperforms the others with scattered missing beats, as well as for $P_{LF}$ with small bursts (Table 7). L gap filling performs better for $P_{LF}$ from 15 s onwards and for $P_{HF}$ with any burst duration. Statistically significant differences are found between all methods at any loss rate. All methods are equally reliable in terms of discrimination for all deletion probabilities and burst durations (Table 8).

NL gap filling remains superior in the Apple Watch dataset in terms of relative error (Table 9), followed by L gap filling. Coverage graphs (Figure 8) show an advantage of NL in $P_{LF}$, while both NL and L gap filling perform similar in $P_{HF}$, although much better than OR. As in simulation, all methods are robust in state discrimination (Figure 9).

### 3.4. Poincaré Plots

As for time-domain metrics, there is no clear difference between correction methods for Poincaré metrics (Table 10).

In the case of scattered missing beats, L performs better with SD1 when the deletion probability is below 25%. There are not significant differences with OR from 25% onwards. NL outperforms the others with SD2, Md and Sd. On the other hand, OR is the best for SD1, SD2 and Md when dealing with bursts. No significant differences can be found with Sd.

Results in terms of group discrimination suggest an advantage of NL gap filling in the case of scattered missing beats, while NL and OR perform similarly when dealing with bursts (Table 11). The three methods perform virtually identically on the Apple Watch dataset, both in terms of relative error (Table 12), coverage (Figure 10) and discrimination (Figure 11). In the last, OR performed better with SD1 and S, in accordance with the simulation.

## 4. Discussion

An analysis of the degradation of some of the most important HRV metrics due to data loss has been presented. A simulation study has been designed to test the influence of missing beats depending on whether they are distributed scattered or in bursts. Correction methods have been tested with both simulation and experimental data, recorded with a commercial wearable. Note that, in contrast to the simulation dataset, PRV was compared to the HRV in the case of the Apple Watch dataset. Thus, the error results obtained for the simulation dataset should not be compared with those obtained for the Apple Watch dataset. Nevertheless, correction methods within the same dataset can still be compared. In the following, a discussion of the best correction method for each metric is given, as well as the maximum acceptable missing beats for a relative error less than 20% in the third quartile. A summary is shown in Table 13.

Regarding time-domain metrics, noticeable differences are only found in the relative error results of the simulation. NL is the best option in case of applications where MHR is the only interesting metric, as it is the best correction method both with bursts and scattered missing beats. NL is also the best-performing method for SDNN with scattered losses. OR is a reliable correction for SDNN and RMSSD in datasets with burst predominance, while RMSSD should be computed using L with scattered missing beats. The robustness of MHR using both L and NL gap filling supports the idea that the number of missing beats is well approximated by these methods. Gap-filling degradation with bursts of missing beats is easily explained by the lack of information as the correction moves away from the edges of the burst. Phenomena such as respiratory sinus arrhythmia also cannot be inferred in large bursts. MHR proved to be a very robust metric in missing data scenarios, assuming a worst-case maximum deviation of 0.7 beats per minute. Although not shown in our results, MHR was able to withstand losses in bursts of up to one minute without the median error exceeding 1 beat per minute. However, it is not easy to establish a threshold for which it is preferable to reject the segment. This will rather depend on the stationarity of the data. Because of the metric’s robustness, in periods where variations are expected, the rate of these changes should be a more dominant factor than metric degradation in the segment rejection decision. The case of scattered losses can be more complex, as depending on the distribution, it can be complicated for an outlier detection method to correctly work. This is magnified in cases with large respiratory sinus arrhythmia oscillations. Segment rejection is encouraged when computing RMSSD with >25% missing beats, as the third quartile error is around 20%. In any case, attempting to correct segments with more than 35% missing beats or a 20 s burst is not considered adequate.

Regarding frequency metrics calculated via FFT, gap-filling methods show a clear advantage in terms of error and state discrimination. NL was the best correction method for datasets with scattered missing beats predominance. In datasets with burst predominance, NL performed better for $P_{LF}$, while L obtained better results for $P_{HF}$. The third quartile of $P_{LF}$ error is greater than 20% in case of segments with more than 25% scattered missing beats, suggesting that those segments should be discarded for $P_{LF}$ analysis. In the case of $P_{HF}$, segments with more than 15% missing beats should also be discarded. Discarding segments is suggested when analyzing missing data in bursts longer than 10 s.

In regards to Lomb’s method, NL obtained the best results for scattered missing beats. In datasets with burst predominance, L obtained the best results for $P_{HF}$, while NL obtained the best results for $P_{LF}$. Segment rejection for $P_{LF}$ analysis is suggested with more than 25% scattered missing beats. In case of $P_{HF}$ analysis, rejection is suggested with more than 15% missing beats. Segments should be discarded for bursts longer than 10 s as well. Although Lomb’s method allows its use without gap filling—in fact, with no interpolation at all—it deteriorates rapidly in the absence of the whole series (OR case). This is explained due to the phenomenon of the over-oscillation of the spectrum as samples are discarded, whose effect is limited when calculating the power by integrating [39], but still causes a degradation of the metrics.

In the case of Poincaré metrics, NL obtained better results in the case of scattered losses for most metrics, in terms of both error and discrimination between states. L obtained the best results when analyzing specifically SD1 up to 25% of missing beats, while OR obtained better results with more than 25% missing beats. However, the third quartile error is greater than 20% in this case, and segment rejection is suggested. As in the case of time-domain metrics, the criterion for rejecting a segment should prioritize the expected stationarity, given the robustness of the metrics with correction methods. OR obtained the best results in the case of bursts for all metrics.

The proposed gap-filling method, especially in its non-linear version, has been demonstrated to be a very effective correction method. In [24], the difference between correcting the interval series, as is the case with most of the methods in the literature, and correcting the event series, i.e., the beat-occurrence timestamps, was shown. Correcting the interval series involves shifting the timestamps of subsequent beats to address the interval correction. This ultimately means forgoing the reference provided by the subsequent, well-detected beats. Instead, the proposed method corrects the event series without this shifting by adding a variable number of beats, taking into account the budget of seconds to be filled in. Larger gaps require a greater number of filling beats to obtain IBIs in accordance with the adjacent intervals to the gap. In [24], it is shown that event correction yields more accurate results than interval series correction. Besides, a novel aspect of the proposed gap-filling method lies in the way in which the correction of each segment is approached. The proposed method is a segment-based iterative algorithm instead of a gap-based one. The use of this kind of algorithm aims to cope with two major problems of event series gap filling: distinguishing outliers at high loss rates and the lack of knowledge of the number of missing beats per gap. Thus, it starts by solving simple gaps before those involving more than one beat. This is an improvement over the majority of gap-filling methods in the literature, where each gap is corrected before moving on to the next one, missing the advantage of solving the shorter gaps first.

It should be noted that the best method is not necessarily the one with the lowest error. Depending on the application, especially working with devices with limited computational capacity and/or which are battery-operated, a method with acceptable results is interesting if it means an improvement in computation time and overall processing load.

### Limitations

Regarding the limitations of this work, it is important to note that this research only focuses on data losses—false negatives in beat detections—and not on general errors—a combination of false positives and false negatives. The presence of false positives has a deleterious impact when trying to obtain the most accurate metrics. This type of error introduces an additional variable: the baseline from which to infer false negatives could be lost. In addition to a previous artifact detection stage, a false beat detection rejection stage should be implemented before applying the presented methods to deal with missing data. If the number of false beat detections is not very high, a moving-average-based algorithm may be enough. This concept is of paramount importance when dealing with wearable devices, especially those that monitor 24/7, since beat detections can be unreliable a high percentage of the time, and therefore for any further processing.

Another limitation is the monotonicity of Hermite polynomials. As this interpolation eliminates relative maxima and minima within the burst, it should be taken into account in cases with long bursts and high variability, such as in cases with strong respiratory sinus arrhythmia. Despite this, it performs better than other traditional interpolation methods in the literature, such as cubic splines, which present convergence problems by introducing unwanted oscillations. Further work should be done to address this, for example, by introducing estimated stationary points before interpolating. In addition, interpolation methods that do not impose monotonicity while limiting overshooting should also be investigated.

In addition, in contrast to the simulation database, respiratory frequencies have not been tested for the Apple Watch database. Therefore, the use of classical frequency bands may result in an incorrect evaluation of the frequency metrics in some cases, and their behavior may differ from that seen in simulation [34]. However, data presentation in medians and quartiles should limit the effect of these outliers in the results.

## 5. Conclusions

A segment-based gap-filling method for HRV series analysis in the presence of missing data has been presented. Correction is made on the event series, allowing this method to be used independently of the signal used for beat detection (ECG, PPG, etc.). The best-performing correction methodology depends on the analyzed HRV metrics: correction without gap filling is the best option for SDNN, RMSSD and Poincaré plot metrics in situations when the missing beats are mainly in bursts, whereas they benefit from gap-filling approaches in the cases of scattered missing beats. Gap-filling approaches obtained the best performance in terms of frequency-domain metrics. Furthermore, the performance analysis allows us to extract some conclusions about when to discard a segment for further analysis depending on how much error is assumable in the specific application: in order to obtain estimations with an error lower than 20%, those segments with more than 35% of missing beats or more than 20 s bursts should be discarded for HRV time-domain metrics and Poincaré plots. Moreover, those segments with more than 25% of missing beats or more than 10 s bursts should be discarded for HRV frequency-domain analysis.

## Figures and Tables

**Figure 1 sensors-22-05774-f001:**
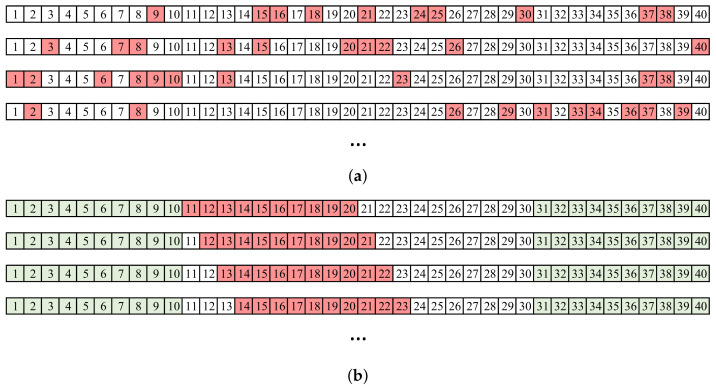
Example of simulation with a segment of 40 beats. Deleted beats are displayed in red. (**a**) Random distributed missing beats, p=0.25. (**b**) Bursts of missing beats. The elements at the ends (green) cannot be deleted.

**Figure 2 sensors-22-05774-f002:**
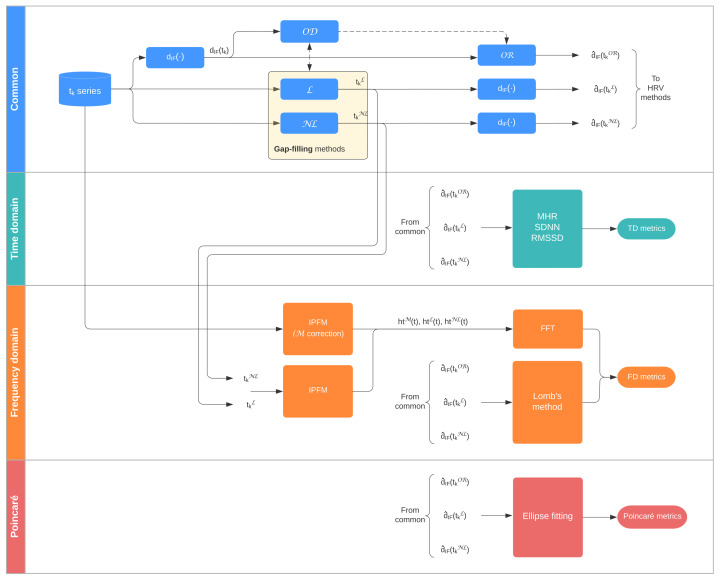
Process flow. OD = Outlier Detection; OR = Outlier Rejection; L = Linear; NL = Non-Linear; M = Model-based.

**Figure 3 sensors-22-05774-f003:**
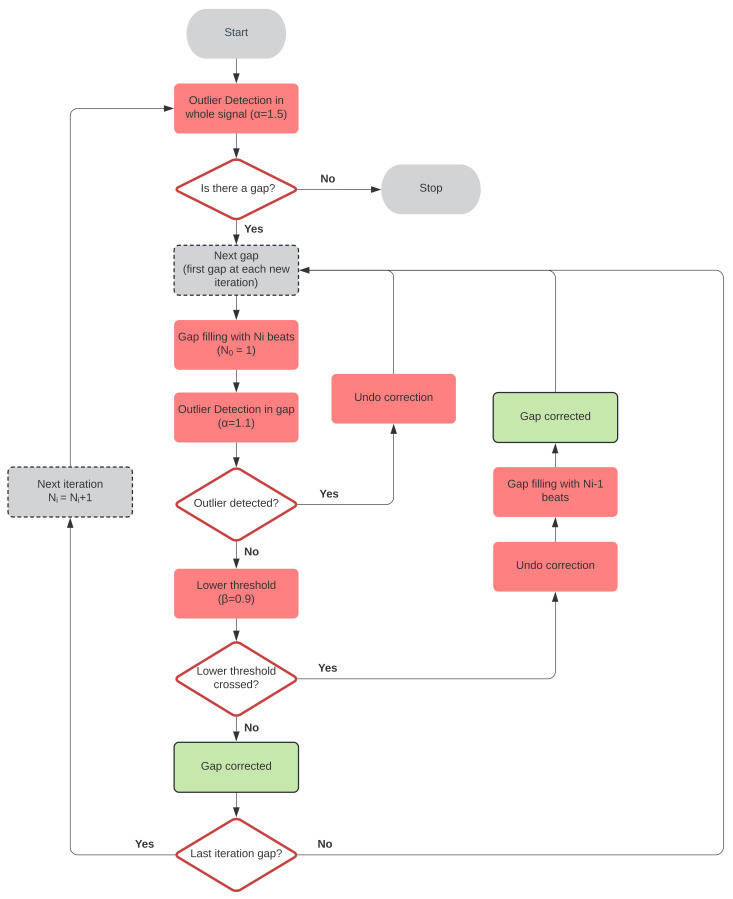
Gap-filling algorithm flowchart.

**Figure 4 sensors-22-05774-f004:**
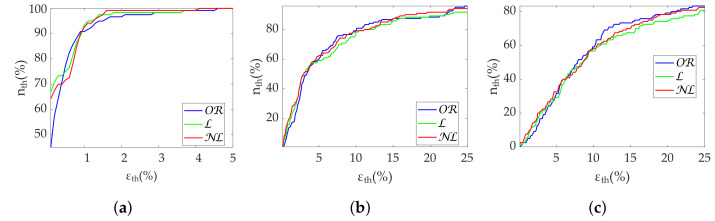
**Coverage of time-domain metrics from Apple Watch dataset.** (**a**) MHR. (**b**) SDNN. (**c**) RMSSD.

**Figure 5 sensors-22-05774-f005:**
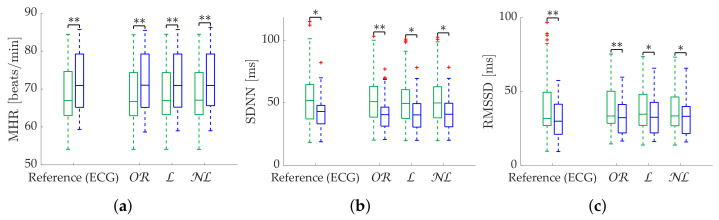
**Relax (green)/stress (blue) discrimination of time-domain metrics from Apple Watch dataset.** (**a**) MHR. (**b**) SDNN. (**c**) RMSSD. *: Significant differences (p<0.05) between relax and stress groups. **: Significant differences (p<0.001) between relax and stress groups.

**Figure 6 sensors-22-05774-f006:**
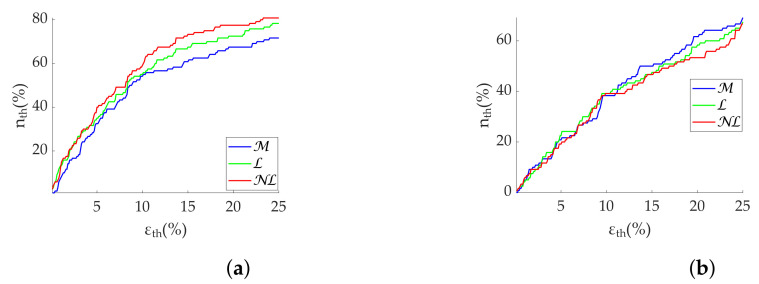
**Coverage of frequency-domain metrics computed via FFT from Apple Watch dataset.** (**a**) $P_{LF}$. (**b**) $P_{HF}$.

**Figure 7 sensors-22-05774-f007:**
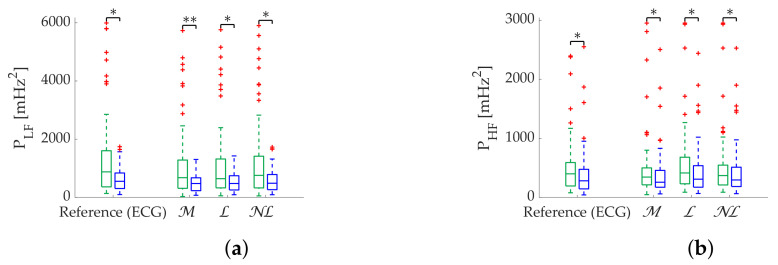
**Relax (green)/Stress (blue) discrimination of frequency-domain metrics computed via FFT from Apple Watch dataset.** (**a**) $P_{LF}$. (**b**) $P_{HF}$. *: Significant differences (p<0.05) between relax and stress groups. **: Significant differences (p<0.001) between relax and stress groups.

**Figure 8 sensors-22-05774-f008:**
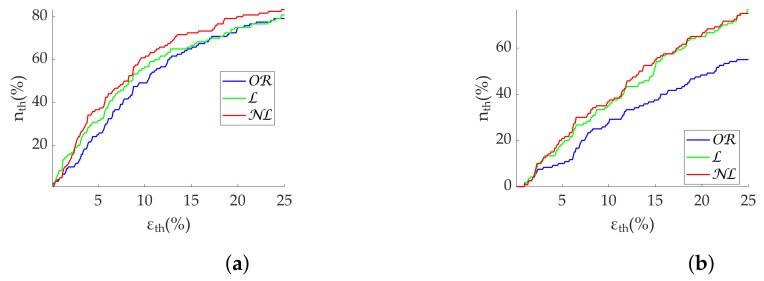
**Coverage of frequency-domain metrics computed via Lomb’s method from Apple Watch dataset.** (**a**) $P_{LF}$. (**b**) $P_{HF}$.

**Figure 9 sensors-22-05774-f009:**
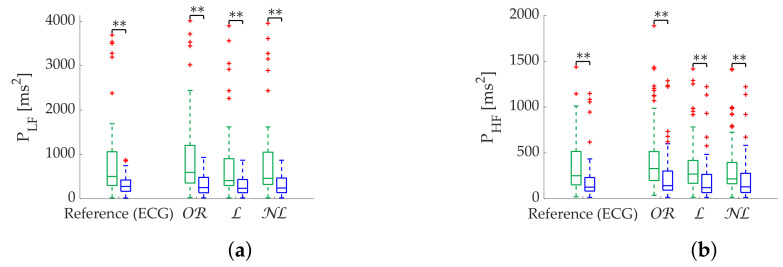
**Relax (green)/Stress (blue) discrimination of frequency-domain metrics computed via Lomb’s method from Apple Watch dataset.** (**a**) $P_{LF}$. (**b**) $P_{HF}$. **: Significant differences (p<0.001) between relax and stress groups.

**Figure 10 sensors-22-05774-f010:**
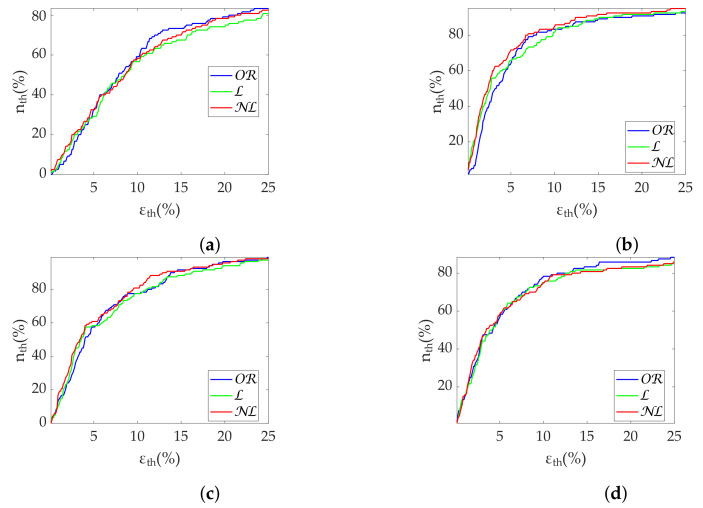
**Coverage of Poincaré metrics from Apple Watch dataset.** (**a**) SD1. (**b**) SD2. (**c**) Md. (**d**) Sd.

**Figure 11 sensors-22-05774-f011:**
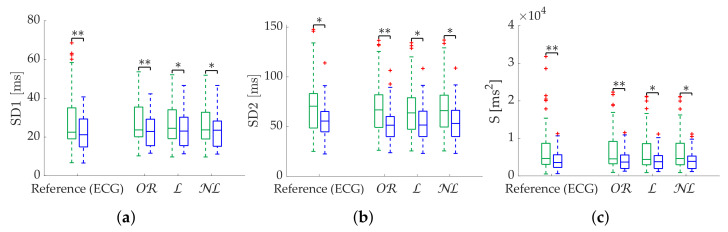
**Relax (green)/stress (blue) discrimination of Poincaré metrics from Apple Watch dataset.** (**a**) SD1. (**b**) SD2. (**c**) S. (**d**) Md. (**e**) Sd. *: Significant differences (p<0.05) between relax and stress groups. **: Significant differences (p<0.001) between relax and stress groups.

**Table 1 sensors-22-05774-t001:** **Relative error (%) of time-domain metrics. (a)** Scattered missing beats. **(b)** Bursts. †: Significant differences (p<0.05) between OR and L. △: Significant differences (p<0.05) between L and NL. §: Significant differences (p<0.05) between NL and OR.

(a)					
**Method**	**Metric**	**Deletion Probability (%)**			
		**5**	**15**	**25**	**35**
OR	MHR	0.13 (0.05–0.24)	0.25 (0.12–0.48) ${}^{†}$	0.39 (0.20–0.74) ${}^{†}$	0.54 (0.28–1.02) ${}^{†}$
	SDNN	1.80 (0.85–3.07) ${}^{†}$	3.61 (1.76–6.03) ${}^{†}$	5.10 (2.34–9.18) ${}^{†}$	7.32 (3.11–14.93) ${}^{†}$
	RMSSD	2.09 (0.95–4.03) ${}^{†}$	5.40 (2.12–9.23) ${}^{†}$	8.90 (3.96–14.55)	10.84 (5.21–23.97)
L	MHR	0.00 (0.00–0.01) ${}^{▵}$	0.01 (0.01–0.03) ${}^{▵}$	0.03 (0.01–0.49) ${}^{▵}$	0.08 (0.03–0.75) ${}^{▵}$
	SDNN	0.43 (0.19–0.81) ${}^{▵}$	1.85 (0.81–3.37) ${}^{▵}$	4.71 (2.59–8.25) ${}^{▵}$	8.09 (4.18–14.48) ${}^{▵}$
	RMSSD	1.07 (0.41–2.05) ${}^{▵}$	2.68 (1.14–6.09) ${}^{▵}$	7.90 (2.72–19.56) ${}^{▵}$	13.98 (5.21–37.84) ${}^{▵}$
NL	MHR	0.00 (0.00–0.00)	0.00 (0.00–0.02)	0.02 (0.01–0.09)	0.05 (0.01–0.70) ${}^{§}$
	SDNN	0.16 (0.05–0.44) ${}^{§}$	0.77 (0.24–2.34) ${}^{§}$	2.63 (0.85–6.00) ${}^{§}$	5.42 (2.10–10.18) ${}^{§}$
	RMSSD	1.13 (0.44–2.28) ${}^{§}$	4.14 (2.13–7.43) ${}^{§}$	9.42 (4.95–16.70) ${}^{§}$	14.50 (7.54–29.35) ${}^{§}$
**(b)**					
**Method**	**Metric**	**Burst duration (s)**			
		**5**	**10**	**15**	**20**
OR	MHR	0.16 (0.07–0.28)	0.22 (0.11–0.44) ${}^{†}$	0.31 (0.14–0.56) ${}^{†}$	0.40 (0.17–0.71) ${}^{†}$
	SDNN	1.69 (0.84–2.43) ${}^{†}$	2.12 (1.10–3.65) ${}^{†}$	3.06 (1.40–4.87) ${}^{†}$	3.55 (1.72–5.96) ${}^{†}$
	RMSSD	1.66 (0.83–2.44) ${}^{†}$	2.43 (1.14–3.74) ${}^{†}$	3.15 (1.58–5.38) ${}^{†}$	4.08 (2.07–7.08) ${}^{†}$
L	MHR	0.01 (0.00–0.03) ${}^{▵}$	0.03 (0.01–0.53) ${}^{▵}$	0.47 (0.02–0.76) ${}^{▵}$	0.73 (0.09–0.98) ${}^{▵}$
	SDNN	1.39 (0.57–2.66) ${}^{▵}$	3.41 (1.51–5.32) ${}^{▵}$	4.83 (2.63–7.82) ${}^{▵}$	6.38 (3.14–9.87) ${}^{▵}$
	RMSSD	1.66 (0.75–3.38) ${}^{▵}$	3.60 (1.83–6.23) ${}^{▵}$	4.87 (2.84–8.41) ${}^{▵}$	6.97 (3.95–10.86) ${}^{▵}$
NL	MHR	0.01 (0.00–0.09) ${}^{§}$	0.02 (0.01–0.57) ${}^{§}$	0.09 (0.01–0.72)	0.55 (0.03–0.83)
	SDNN	1.24 (0.44–3.02) ${}^{§}$	2.84 (1.04–4.82) ${}^{§}$	4.46 (2.15–7.02) ${}^{§}$	5.80 (2.86–8.85) ${}^{§}$
	RMSSD	1.77 (0.90–3.87) ${}^{§}$	3.77 (2.13–6.51) ${}^{§}$	5.80 (3.40–9.01) ${}^{§}$	7.78 (4.36–12.17) ${}^{§}$

**Table 2 sensors-22-05774-t002:** ***p*****-values of ranked signed test for supine/tilt discrimination of time-domain metrics.** N.S.: Not significant (p>0.05).

Method	Metric	Reference		Deletion Probability (%)					Burst Duration (s)			
				5	15	25	35		5	10	15	20
OR	MHR	<$10^{-3}$		<$10^{-3}$	<$10^{-3}$	<$10^{-3}$	<$10^{-3}$		<$10^{-3}$	<$10^{-3}$	<$10^{-3}$	<$10^{-3}$
	SDNN	0.020		0.011	0.016	N.S.	N.S.		0.011	0.011	0.011	0.011
	RMSSD	<$10^{-3}$		<$10^{-3}$	<$10^{-3}$	<$10^{-3}$	<$10^{-3}$		<$10^{-3}$	<$10^{-3}$	<$10^{-3}$	<$10^{-3}$
L	MHR	<$10^{-3}$		<$10^{-3}$	<$10^{-3}$	<$10^{-3}$	<$10^{-3}$		<$10^{-3}$	<$10^{-3}$	<$10^{-3}$	<$10^{-3}$
	SDNN	0.020		0.014	0.008	N.S.	N.S.		0.011	0.011	0.011	0.004
	RMSSD	<$10^{-3}$		<$10^{-3}$	<$10^{-3}$	<$10^{-3}$	<$10^{-3}$		<$10^{-3}$	<$10^{-3}$	<$10^{-3}$	<$10^{-3}$
NL	MHR	<$10^{-3}$		<$10^{-3}$	<$10^{-3}$	<$10^{-3}$	<$10^{-3}$		<$10^{-3}$	<$10^{-3}$	<$10^{-3}$	<$10^{-3}$
	SDNN	0.020		0.012	0.011	0.021	0.014		0.014	0.007	0.012	0.026
	RMSSD	<$10^{-3}$		<$10^{-3}$	<$10^{-3}$	<$10^{-3}$	<$10^{-3}$		<$10^{-3}$	<$10^{-3}$	<$10^{-3}$	<$10^{-3}$

**Table 3 sensors-22-05774-t003:** **Relative error (%) of time-domain metrics from Apple Watch dataset.** †: Significant differences (p<0.05) between OR and L. △: Significant differences (p<0.05) between L and NL. §: Significant differences (p<0.05) between NL and OR.

Metric	Method		
	OR	L	NL
MHR	0.12 (0.04–0.47)	0.03 (0.01–0.52) ${}^{▵}$	0.03 (0.01–0.66) ${}^{§}$
SDNN	3.36 (1.97–7.47) ${}^{†}$	2.92 (1.51–9.55) ${}^{▵}$	2.96 (1.31–8.62)
RMSSD	7.84 (4.29–15.90) ${}^{†}$	8.56 (3.99–20.22) ${}^{▵}$	8.61 (3.74–17.69) ${}^{§}$

**Table 4 sensors-22-05774-t004:** **Relative error (%) of frequency-domain metrics computed via FFT. (a)** Scattered missing beats. **(b)** Bursts. †: Significant differences (p<0.05) between M and L. △: Significant differences (p<0.05) between L and NL. §: Significant differences (p<0.05) between NL and M.

(a)					
**Method**	**Metric**	**Deletion Probability (%)**			
		**5**	**15**	**25**	**35**
M	$P_{LF}$	8.08 (3.46–19.10) ${}^{†}$	20.29 (11.37–32.72) ${}^{†}$	36.36 (22.36–53.64) ${}^{†}$	55.16 (29.10–161.86) ${}^{†}$
	$P_{HF}$	15.37 (7.00–30.10) ${}^{†}$	32.89 (20.17–45.24) ${}^{†}$	50.12 (36.46–63.16) ${}^{†}$	59.41 (42.65–73.21) ${}^{†}$
L	$P_{LF}$	0.99 (0.39–2.34) ${}^{▵}$	4.28 (2.04–9.24) ${}^{▵}$	10.94 (5.66–18.71) ${}^{▵}$	15.98 (8.81–28.74) ${}^{▵}$
	$P_{HF}$	2.81 (1.19–5.47) ${}^{▵}$	10.83 (5.54–17.64) ${}^{▵}$	22.69 (11.98–41.40) ${}^{▵}$	34.04 (19.54–61.20)
NL	$P_{LF}$	0.41 (0.15–1.11) ${}^{§}$	1.44 (0.48–4.71) ${}^{§}$	4.24 (1.41–12.57) ${}^{§}$	8.96 (2.20–21.22) ${}^{§}$
	$P_{HF}$	1.63 (0.71–4.16) ${}^{§}$	6.88 (2.45–14.99) ${}^{§}$	18.97 (9.80–37.55) ${}^{§}$	29.20 (17.06–54.77) ${}^{§}$
**(b)**					
**Method**	**Metric**	**Burst duration (s)**			
		**5**	**10**	**15**	**20**
M	$P_{LF}$	10.20 (3.53–20.75) ${}^{†}$	14.62 (6.52–26.29) ${}^{†}$	21.90 (9.95–32.57) ${}^{†}$	26.50 (14.26–39.04) ${}^{†}$
	$P_{HF}$	12.62 (6.38–28.40) ${}^{†}$	17.99 (9.32–32.90) ${}^{†}$	22.82 (14.13–36.28) ${}^{†}$	28.65 (18.53–43.37) ${}^{†}$
L	$P_{LF}$	4.94 (1.70–12.34) ${}^{▵}$	10.89 (4.67–19.12) ${}^{▵}$	15.45 (7.13–26.16) ${}^{▵}$	19.25 (8.88–31.24)
	$P_{HF}$	6.81 (2.92–11.42) ${}^{▵}$	9.95 (4.98–17.20) ${}^{▵}$	14.02 (7.06–22.67) ${}^{▵}$	18.26 (9.41–28.11) ${}^{▵}$
NL	$P_{LF}$	4.72 (1.56–12.18) ${}^{§}$	10.01 (4.34–17.88) ${}^{§}$	13.31 (6.35–25.36) ${}^{§}$	19.34 (9.28–30.70) ${}^{§}$
	$P_{HF}$	6.82 (3.36–11.76) ${}^{§}$	11.02 (5.73–17.59) ${}^{§}$	15.19 (7.85–23.70) ${}^{§}$	19.10 (10.94–29.80) ${}^{§}$

**Table 5 sensors-22-05774-t005:** ***p*****-values of ranked signed test for supine/tilt discrimination of frequency-domain metrics computed via FFT.** N.S.: Not significant (p>0.05).

Method	Metric	Reference		Deletion Probability (%)					Burst Duration (s)			
				5	15	25	35		5	10	15	20
M	$P_{HF}$	<$10^{-3}$		<$10^{-3}$	<$10^{-3}$	<$10^{-3}$	N.S.		<$10^{-3}$	<$10^{-3}$	<$10^{-3}$	<$10^{-3}$
	$P_{LFn}$	<$10^{-3}$		<$10^{-3}$	<$10^{-3}$	<$10^{-3}$	<$10^{-3}$		<$10^{-3}$	<$10^{-3}$	<$10^{-3}$	<$10^{-3}$
	$P_{LF}$/$P_{HF}$	<$10^{-3}$		<$10^{-3}$	<$10^{-3}$	<$10^{-3}$	0.005		<$10^{-3}$	<$10^{-3}$	<$10^{-3}$	<$10^{-3}$
L	$P_{HF}$	<$10^{-3}$		<$10^{-3}$	<$10^{-3}$	<$10^{-3}$	<$10^{-3}$		<$10^{-3}$	<$10^{-3}$	<$10^{-3}$	<$10^{-3}$
	$P_{LFn}$	<$10^{-3}$		<$10^{-3}$	<$10^{-3}$	<$10^{-3}$	<$10^{-3}$		<$10^{-3}$	<$10^{-3}$	<$10^{-3}$	<$10^{-3}$
	$P_{LF}$/$P_{HF}$	<$10^{-3}$		<$10^{-3}$	<$10^{-3}$	<$10^{-3}$	<$10^{-3}$		<$10^{-3}$	<$10^{-3}$	<$10^{-3}$	<$10^{-3}$
NL	$P_{HF}$	<$10^{-3}$		<$10^{-3}$	<$10^{-3}$	<$10^{-3}$	<$10^{-3}$		<$10^{-3}$	<$10^{-3}$	<$10^{-3}$	<$10^{-3}$
	$P_{LFn}$	<$10^{-3}$		<$10^{-3}$	<$10^{-3}$	<$10^{-3}$	<$10^{-3}$		<$10^{-3}$	<$10^{-3}$	<$10^{-3}$	<$10^{-3}$
	$P_{LF}$/$P_{HF}$	<$10^{-3}$		<$10^{-3}$	<$10^{-3}$	<$10^{-3}$	<$10^{-3}$		<$10^{-3}$	<$10^{-3}$	<$10^{-3}$	<$10^{-3}$

**Table 6 sensors-22-05774-t006:** **Relative error (%) of frequency-domain metrics computed via FFT from Apple Watch dataset.** †: Significant differences (p<0.05) between M and L. △: Significant differences (p<0.05) between L and NL. §: Significant differences (p<0.05) between NL and M.

Metric	Method		
	M	L	NL
$P_{LF}$	0.09 (0.04–0.30) ${}^{†}$	0.08 (0.03–0.22) ${}^{▵}$	0.08 (0.03–0.17) ${}^{§}$
$P_{HF}$	0.14 (0.07–0.31) ${}^{†}$	0.16 (0.07–0.30) ${}^{▵}$	0.17 (0.07–0.31) ${}^{§}$

**Table 7 sensors-22-05774-t007:** **Relative error (%) of frequency-domain metrics computed via Lomb’s method. (a)** Scattered missing beats. **(b)** Bursts. †: Significant differences (p<0.05) between OR and L. △: Significant differences (p<0.05) between L and NL. §: Significant differences (p<0.05) between NL and OR.

(a)					
**Method**	**Metric**	**Deletion Probability (%)**			
		**5**	**15**	**25**	**35**
OR	$P_{LF}$	10.75 (4.77–18.84) ${}^{†}$	23.45 (9.90–40.99) ${}^{†}$	34.71 (17.49–66.27) ${}^{†}$	58.11 (24.93–123.23) ${}^{†}$
	$P_{HF}$	23.01 (11.38–45.74) ${}^{†}$	79.42 (46.93–155.29) ${}^{†}$	160.28 (87.39–296.90) ${}^{†}$	304.57 (142.89–665.16) ${}^{†}$
L	$P_{LF}$	0.89 (0.35–1.90) ${}^{▵}$	3.75 (1.60–7.49) ${}^{▵}$	9.90 (4.56–18.19) ${}^{▵}$	15.77 (7.56–28.59) ${}^{▵}$
	$P_{HF}$	2.62 (1.10–4.81) ${}^{▵}$	8.94 (4.93–16.22) ${}^{▵}$	21.27 (11.03–37.15) ${}^{▵}$	30.78 (17.23–61.62) ${}^{▵}$
NL	$P_{LF}$	0.37 (0.13–1.06) ${}^{§}$	1.36 (0.44–3.95) ${}^{§}$	3.43 (1.17–11.70) ${}^{§}$	7.58 (2.28–22.67) ${}^{§}$
	$P_{HF}$	1.45 (0.51–3.31) ${}^{§}$	5.46 (1.92–12.01) ${}^{§}$	16.22 (8.12–31.57) ${}^{§}$	28.33 (14.72–52.65) ${}^{§}$
**(b)**					
**Method**	**Metric**	**Burst Duration (s)**			
		**5**	**10**	**15**	**20**
OR	$P_{LF}$	11.19 (6.69–17.18) ${}^{†}$	18.66 (10.87–28.82) ${}^{†}$	25.33 (12.89–38.36) ${}^{†}$	29.23 (14.57–48.91) ${}^{†}$
	$P_{HF}$	14.06 (7.55–19.79) ${}^{†}$	22.86 (12.70–34.06) ${}^{†}$	30.88 (18.39–45.27) ${}^{†}$	39.17 (24.01–60.79) ${}^{†}$
L	$P_{LF}$	4.48 (1.65–11.24) ${}^{▵}$	9.99 (3.64–19.00) ${}^{▵}$	13.85 (6.53–23.87) ${}^{▵}$	17.38 (7.14–28.54) ${}^{▵}$
	$P_{HF}$	5.51 (2.26–11.05) ${}^{▵}$	8.74 (3.94–18.11) ${}^{▵}$	13.18 (5.10–21.58) ${}^{▵}$	16.22 (7.42–26.31) ${}^{▵}$
NL	$P_{LF}$	4.58 (1.68–11.53) ${}^{§}$	8.43 (3.55–17.21) ${}^{§}$	13.49 (5.93–23.56) ${}^{§}$	18.41 (9.23–28.86) ${}^{§}$
	$P_{HF}$	6.00 (2.79–11.69) ${}^{§}$	10.92 (5.24–19.42) ${}^{§}$	14.60 (7.65–23.30) ${}^{§}$	18.46 (9.45–29.31) ${}^{§}$

**Table 8 sensors-22-05774-t008:** ***p*****-values of ranked signed test for supine/tilt discrimination of frequency-domain metrics computed via Lomb’s method.** N.S.: Not significant (p>0.05).

Method	Metric	Reference		Deletion Probability (%)					Burst Duration (s)			
				5	15	25	35		5	10	15	20
OR	$P_{HF}$	<$10^{-3}$		<$10^{-3}$	<$10^{-3}$	<$10^{-3}$	<$10^{-3}$		<$10^{-3}$	<$10^{-3}$	<$10^{-3}$	<$10^{-3}$
	$P_{LFn}$	<$10^{-3}$		<$10^{-3}$	<$10^{-3}$	<$10^{-3}$	<$10^{-3}$		<$10^{-3}$	<$10^{-3}$	<$10^{-3}$	<$10^{-3}$
	$P_{LF}$/$P_{HF}$	<$10^{-3}$		<$10^{-3}$	<$10^{-3}$	<$10^{-3}$	<$10^{-3}$		<$10^{-3}$	<$10^{-3}$	<$10^{-3}$	<$10^{-3}$
L	$P_{HF}$	<$10^{-3}$		<$10^{-3}$	<$10^{-3}$	<$10^{-3}$	<$10^{-3}$		<$10^{-3}$	<$10^{-3}$	<$10^{-3}$	<$10^{-3}$
	$P_{LFn}$	<$10^{-3}$		<$10^{-3}$	<$10^{-3}$	<$10^{-3}$	<$10^{-3}$		<$10^{-3}$	<$10^{-3}$	<$10^{-3}$	<$10^{-3}$
	$P_{LF}$/$P_{HF}$	<$10^{-3}$		<$10^{-3}$	<$10^{-3}$	<$10^{-3}$	<$10^{-3}$		<$10^{-3}$	<$10^{-3}$	<$10^{-3}$	<$10^{-3}$
NL	$P_{HF}$	<$10^{-3}$		<$10^{-3}$	<$10^{-3}$	<$10^{-3}$	<$10^{-3}$		<$10^{-3}$	<$10^{-3}$	<$10^{-3}$	<$10^{-3}$
	$P_{LFn}$	<$10^{-3}$		<$10^{-3}$	<$10^{-3}$	<$10^{-3}$	<$10^{-3}$		<$10^{-3}$	<$10^{-3}$	<$10^{-3}$	<$10^{-3}$
	$P_{LF}$/$P_{HF}$	<$10^{-3}$		<$10^{-3}$	<$10^{-3}$	<$10^{-3}$	<$10^{-3}$		<$10^{-3}$	<$10^{-3}$	<$10^{-3}$	<$10^{-3}$

**Table 9 sensors-22-05774-t009:** **Relative error (%) of frequency-domain computed via Lomb’s method metrics from Apple Watch dataset.** †: Significant differences (p<0.05) between OR and L. △: Significant differences (p<0.05) between L and NL. §: Significant differences (p<0.05) between NL and OR.

Metric	Method		
	OR	L	NL
$P_{LF}$	0.10 (0.05–0.24) ${}^{†}$	0.08 (0.03–0.23) ${}^{▵}$	0.08 (0.03–0.18) ${}^{§}$
$P_{HF}$	0.20 (0.08–0.56) ${}^{†}$	0.15 (0.06–0.32) ${}^{▵}$	0.13 (0.06–0.25) ${}^{§}$

**Table 10 sensors-22-05774-t010:** **Relative error (%) of Poincaré metrics. (a)** Scattered missing beats. **(b)** Bursts. †: Significant differences (p<0.05) between OR and L. △: Significant differences (p<0.05) between L and NL. §: Significant differences (p<0.05) between NL and OR.

(a)					
**Method**	**Metric**	**Deletion Probability (%)**			
		**5**	**15**	**25**	**35**
OR	SD1	2.13 (0.94–3.96) ${}^{†}$	5.33 (2.24–9.25) ${}^{†}$	9.06 (4.10–14.36)	10.66 (5.29–23.67)
	SD2	2.58 (1.31–4.28) ${}^{†}$	4.93 (2.04–8.53) ${}^{†}$	7.20 (3.22–13.05) ${}^{†}$	10.75 (4.93–20.32) ${}^{†}$
	Md	2.40 (1.16–4.15) ${}^{†}$	4.49 (2.18–7.44) ${}^{†}$	6.35 (2.87–10.78) ${}^{†}$	8.70 (4.10–17.82) ${}^{†}$
	Sd	2.80 (1.31–4.95) ${}^{†}$	6.19 (2.87–11.14)	10.05 (4.47–18.67) ${}^{†}$	14.46 (7.16–31.01) ${}^{†}$
L	SD1	1.07 (0.41–2.05) ${}^{▵}$	2.68 (1.14–6.09) ${}^{▵}$	7.90 (2.72–19.56) ${}^{▵}$	14.00 (5.21–37.87) ${}^{▵}$
	SD2	0.40 (0.19–0.93) ${}^{▵}$	1.85 (0.97–3.30) ${}^{▵}$	4.28 (2.24–7.08) ${}^{▵}$	6.99 (3.95–13.22) ${}^{▵}$
	Md	0.52 (0.22–1.08) ${}^{▵}$	2.05 (0.85–4.18) ${}^{▵}$	4.46 (2.07–7.23) ${}^{▵}$	7.11 (3.54–11.93) ${}^{▵}$
	Sd	0.47 (0.17–0.98) ${}^{▵}$	1.34 (0.52–3.23) ${}^{▵}$	3.59 (1.27–12.57) ${}^{▵}$	7.09 (1.78–33.63) ${}^{▵}$
NL	SD1	1.13 (0.44–2.28) ${}^{§}$	4.14 (2.13–7.43) ${}^{§}$	9.42 (4.95–16.70) ${}^{§}$	14.51 (7.54–29.39) ${}^{§}$
	SD2	0.12 (0.04–0.30) ${}^{§}$	0.56 (0.15–1.50) ${}^{§}$	1.67 (0.54–3.98) ${}^{§}$	3.39 (1.19–8.51) ${}^{§}$
	Md	0.23 (0.09–0.56) ${}^{§}$	1.06 (0.27–3.00) ${}^{§}$	2.43 (1.04–5.73) ${}^{§}$	4.83 (1.88–9.33) ${}^{§}$
	Sd	0.30 (0.10–0.76) ${}^{§}$	0.90 (0.26–2.46) ${}^{§}$	2.39 (0.77–7.89) ${}^{§}$	4.92 (1.60–17.24) ${}^{§}$
**Method**	**Metric**	**Burst Duration (s)**			
		**5**	**10**	**15**	**20**
OR	SD1	1.69 (0.84–2.50) ${}^{†}$	2.45 (1.16–3.76) ${}^{†}$	3.19 (1.63–5.39) ${}^{†}$	4.03 (2.00–7.04) ${}^{†}$
	SD2	1.85 (0.89–2.74) ${}^{†}$	2.44 (1.29–3.92) ${}^{†}$	3.24 (1.57–5.11) ${}^{†}$	3.94 (1.89–6.39) ${}^{†}$
	Md	1.91 (0.94–3.05) ${}^{†}$	2.54 (1.04–4.59) ${}^{†}$	3.49 (1.26–5.86) ${}^{†}$	4.14 (2.01–7.40) ${}^{†}$
	Sd	1.50 (0.90–2.49)	2.32 (1.21–3.65)	2.98 (1.65–4.54)	3.57 (2.02–5.67)${}^{†}$
L	SD1	1.67 (0.75–3.38) ${}^{▵}$	3.60 (1.83–6.23) ${}^{▵}$	4.88 (2.84–8.41) ${}^{▵}$	6.97 (3.95–10.86) ${}^{▵}$
	SD2	1.31 (0.56–2.70) ${}^{▵}$	3.40 (1.36–5.37) ${}^{▵}$	4.90 (2.49–7.88) ${}^{▵}$	6.26 (3.10–10.14) ${}^{▵}$
	Md	2.33 (0.97–4.11) ${}^{▵}$	5.65 (2.90–8.72) ${}^{▵}$	8.44 (4.31–11.97) ${}^{▵}$	10.53 (4.53–14.29) ${}^{▵}$
	Sd	1.97 (0.75–4.06) ${}^{▵}$	3.24 (1.57–6.84)	4.08 (1.92–7.50)	4.68 (2.37–8.05) ${}^{▵}$
NL	SD1	1.77 (0.90–3.87) ${}^{§}$	3.78 (2.13–6.52) ${}^{§}$	5.80 (3.41–9.02) ${}^{§}$	7.78 (4.36–12.17) ${}^{§}$
	SD2	1.16 (0.32–2.87) ${}^{§}$	2.53 (0.89–4.96) ${}^{§}$	4.42 (2.18–6.88) ${}^{§}$	5.60 (2.61–8.70) ${}^{§}$
	Md	1.16 (0.32–2.87) ${}^{§}$	2.53 (0.89–4.96) ${}^{§}$	4.42 (2.18–6.88) ${}^{§}$	5.60 (2.61–8.70) ${}^{§}$
	Sd	1.81 (0.61–4.33)	2.97 (1.21–5.62)	3.53 (1.67–7.63)	4.17 (1.99–7.03)

**Table 11 sensors-22-05774-t011:** ***p*****-values of ranked signed test for supine/tilt discrimination of Poincaré metrics.** N.S.: Not Significative (p>0.05).

Method	Metric	Reference		Deletion Probability (%)					Burst Duration (s)			
				5	15	25	35		5	10	15	20
OR	SD1	<$10^{-3}$		<$10^{-3}$	<$10^{-3}$	<$10^{-3}$	<$10^{-3}$		<$10^{-3}$	<$10^{-3}$	<$10^{-3}$	<$10^{-3}$
	SD2	0.031		N.S.	N.S.	N.S.	N.S.		N.S.	N.S.	N.S.	N.S.
	SD12	<$10^{-3}$		<$10^{-3}$	<$10^{-3}$	<$10^{-3}$	<$10^{-3}$		<$10^{-3}$	<$10^{-3}$	<$10^{-3}$	<$10^{-3}$
	S	<$10^{-3}$		<$10^{-3}$	<$10^{-3}$	0.012	0.002		<$10^{-3}$	<$10^{-3}$	<$10^{-3}$	<$10^{-3}$
	Md	<$10^{-3}$		0.001	0.002	0.155	0.016		0.001	0.002	0.002	0.002
	Sd	0.039		0.009	N.S.	N.S.	N.S.		0.024	0.026	0.022	0.015
L	SD1	<$10^{-3}$		<$10^{-3}$	<$10^{-3}$	<$10^{-3}$	<$10^{-3}$		<$10^{-3}$	<$10^{-3}$	<$10^{-3}$	<$10^{-3}$
	SD2	0.031		N.S.	N.S.	N.S.	N.S.		N.S.	N.S.	N.S.	N.S.
	SD12	<$10^{-3}$		<$10^{-3}$	<$10^{-3}$	<$10^{-3}$	<$10^{-3}$		<$10^{-3}$	<$10^{-3}$	<$10^{-3}$	<$10^{-3}$
	S	<$10^{-3}$		<$10^{-3}$	<$10^{-3}$	<$10^{-3}$	0.007		<$10^{-3}$	<$10^{-3}$	<$10^{-3}$	<$10^{-3}$
	Md	<$10^{-3}$		<$10^{-3}$	0.001	0.010	0.012		<$10^{-3}$	<$10^{-3}$	0.002	0.002
	Sd	0.039		0.034	N.S.	N.S.	N.S.		N.S.	N.S.	N.S.	N.S.
NL	SD1	<$10^{-3}$		<$10^{-3}$	<$10^{-3}$	<$10^{-3}$	<$10^{-3}$		<$10^{-3}$	<$10^{-3}$	<$10^{-3}$	<$10^{-3}$
	SD2	0.031		0.043	N.S.	N.S.	N.S.		N.S.	N.S.	N.S.	N.S.
	SD12	<$10^{-3}$		<$10^{-3}$	<$10^{-3}$	<$10^{-3}$	<$10^{-3}$		<$10^{-3}$	<$10^{-3}$	<$10^{-3}$	<$10^{-3}$
	S	<$10^{-3}$		<$10^{-3}$	<$10^{-3}$	<$10^{-3}$	0.004		<$10^{-3}$	<$10^{-3}$	<$10^{-3}$	<$10^{-3}$
	Md	<$10^{-3}$		<$10^{-3}$	<$10^{-3}$	0.002	0.002		0.002	0.001	0.002	0.013
	Sd	0.039		0.033	N.S.	N.S.	N.S.		0.041	0.041	0.027	0.049

**Table 12 sensors-22-05774-t012:** **Relative error (%) of Poincaré metrics from Apple Watch dataset.** †: Significant differences (p<0.05) between OR and L. △: Significant differences (p<0.05) between L and NL. §: Significant differences (p<0.05) between NL and OR.

Metric	Method		
	OR	L	NL
SD1	7.83 (4.15–15.91) ${}^{†}$	8.56 (3.99–20.22) ${}^{▵}$	8.61 (3.73–17.70) ${}^{§}$
SD2	3.22 (1.59–6.38) ${}^{†}$	2.61 (1.06–7.98) ${}^{▵}$	2.35 (1.06–6.08)
Md	3.97 (2.21–8.28) ${}^{†}$	3.55 (2.03–9.24) ${}^{▵}$	3.40 (1.67–8.42)
Sd	4.20 (1.66–9.42)	3.96 (1.87–10.00) ${}^{▵}$	3.44 (1.49–10.06) ${}^{§}$

**Table 13 sensors-22-05774-t013:** Summary of findings. **(a)** Best correction method. **(b)** Maximum acceptable missing beats for a relative error less than 20% in the third quartile.

(a)		
**Metric**	**Scattered Missing Beats**	**Bursts**
MHR	NL	NL
SDNN	NL	OR
RMSSD	L	OR
$P_{LF}$ (FFT)	NL	NL
$P_{HF}$ (FFT)	NL	L
$P_{LF}$ (Lomb)	NL	NL
$P_{HF}$ (Lomb)	NL	L
SD1	L	OR
SD2	NL	OR
Md	NL	OR
Sd	NL	OR/NL
**(b)**		
**Metric**	**Scattered Missing Beats**	**Bursts**
MHR	35%	20 s
SDNN	35%	20 s
RMSSD	25%	20 s
$P_{LF}$ (FFT)	25%	10 s
$P_{HF}$ (FFT)	15%	10 s
$P_{LF}$ (Lomb)	25%	10 s
$P_{HF}$ (Lomb)	15%	10 s
SD1	25%	20 s
SD2	35%	20 s
Md	35%	20 s
Sd	35%	20 s

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
