# Peer review of "Effects of Missing Data on Heart Rate Variability Metrics"

_sensors, 2022, doi:10.3390/s22155774_

Round 1

Reviewer 1 Report

The paper evaluated the methods to alleviate the effect of data loss in ECG on the HRV analysis. It may attract researchers in the HRV. The results need consolidation for better reading.

Line 2: Does "irruption" mean "exponential growth"? Please update it accordingly. Same at line 27.

Line 5: Delete "from"

Line 14: The keywords are too general. HRV, ANS, Apple Watch, and Poincare plots are good. 

Line 18: Change "in a non-invasive way" to "noninvasively"

Line 50: change "face" to "presence"

Line 61: change "obtain" to "generate"

Line 133: What is "central minute"?

Line 158: Is it "continuous-time basis" or "discrete time basis"?

Line165: Please explain the rationale of equation 3. Should "<" be ">" for a outlier?

Line 213: Please explain the equation r(t). What is m(t) and T?

Table 1, 3, 4, 6, 7, 9, 10, 11, 12: Please provide statistical significance data to justify if a method is better than the other. The result summary of each table should be consolidated based on the statistical significance data. 

Table 2, 5, 8, 11: What do "*" and "**" mean? Please use real values if possible.

Figure 4, 6, 8, 10: Is "Correct cases" "relative error"? Is "Permitted error" "deletion probability"?

The discussion should be consolidated. Remove the statements that are redundant, i.e. statements already in the results section.

Conclusion should be concise. The table is not appropriate in the conclusion. 

Author Response

We thank the Reviewer for the helpful and constructive comments, which have contributed to the improvement of the manuscript, and we address as follows. Please see the attachment.

Reviewer 2 Report

This is a well-designed study for analysis of the measurement accuracy of some popular HRV metrics (in the time domain, frequency domain and from Poincaré plots) in conditions of data loss of several beat-to-beat intervals (single gaps or bursts of long-duration gaps). Three recognized methods are compared for data restoration and HRV measurement in such adverse conditions, namely Outlier Removal, Linear and Non-Linear correction methods. RR-series data from two sources are used, including simulation of gaps and clinical data recorded with gaps. Statistical measurements and comparisons or results seem correct and discussed in sufficient detail. Figures and Tables are correctly illustrated. The team of authors present a high-level of experience on this topic and the use of terms is mostly correct. Some revision comments are highlighted below.

Revision remarks.

  1. Extensive English style corrections are necessary by a native English speaker, especially in the Abstract and Introduction. .
  2. Abstract: Comment applied as follows:
    1. Ln 8: “bursts, the most common due to motion artifacts” -> This statement is misleading that the study simulates bursts of false positive detections, which are common during motion artifacts.
    2. Ln 9: “pulse detection series” -> Specify the kind of detected pulses. Unclear if these are RR-interval time series or PPG beat interval time series or others. Give information on the input signal for beat-to-beat extraction. Provide size of the database for statistical justification.
    3. Abstract misses most important numerical data which are essential information in each abstract to justify the conclusions!
    4. Ln 11: The metrics RMSSD and SD are used without a definition.  
    5. Ln 11: “outlier removal” concerns some methodological concept, which is not earlier specified in the methodological description.
    6. Ln 12-13: “Best approaches for other time-domain and Poincaré plot metrics are dependent on the distribution and number of missing beats.” -> Conclusion not supported by the results, in overall indefinite without numerical limits and explanations about the studied distributions and beats.
  3. Introduction: The reference list is not up-to-date because references after 2017 are missing (except 2 in 2018, one in 2019). Considering the constantly increasing amount of published papers in the field of improving the diagnostics in electrocardiography, as well as versions of ECG portable devices, which measure, store and diagnose only reduced HRV data, I think that the authors could find sufficient amount of new methods for RR-interval data restoration with respective validation/test performance results. Comparison to such state-of-the-art methods is essential, clearly demonstrating the novelty of this study.
  4. Section “1.2. Research contribution” is out of standards for scientific articles. The content of such section could find place in Conclusions but not in Introduction. Instead the last paragraph of Introduction should define the AIMS, which are NOT properly formulated. Introduction should conclude about the unsolved problems in the field of research, and a brief description about the novelty of the proposed solution (without deep methodological terms), highlighting the expected positive results, benefits and implications for the clinicians or industry.
  5. The title of section “2.1. Simulation” is not informative about the type of the simulation. On could expect ECG data simulaton or noise simulation but it is in fact a kind “Simulation of missing beats” (or other formulation if authors consider appropriate).
  6. Ln 111-138 and Figure 1: Only scenario with missing beats is simulated (corresponding to false negative beat detections) but not the scenario with detected false extra beats (corresponding to false positive beat detections). The first seems to be at low SNR while the second is common in conditions of artifacts. The big question is how the scenario with false extra beats is simulated and if not, why?
  7. Figure 1b: The scenario with missing several beats is easily detectable than a single missing beat. Define why you use exactly this scenario and what are the challenges in such cases.
  8. Figure1: Keep the template style using (a), (b)… in the bottom of each plot. Correct this inconsistency in other figures, where applicable and use the template form for placing figures.
  9. Ln 147: “Apple Watch outputs the event timestamps only when the –internal– PPG allows reliable pulse detection… Thus, the derived pulse-to-pulse series present intermittent gaps.” -> Clarify the statement. Furthermore, specify how the Apple Watch and the reference Polar H7 outputs have been matched as soon as data are stored in independent devices without synchronous recording.
  10. Ln 150: “Missing data represent around the 10% of total events” -> Define more details on the database: What is the total number of events? What is the number of segments with and without gaps and how many beat-to-beat intervals are included in analysis?
  11. Ln 234-235: “As S and SD1/SD2 are computed from SD1 and SD2, value degradation is not shown for these metrics.” -> Incomprehensive. What does it mean “value degradation”? This statements resembles like a conclusion based on some results, which are, however, missing. Methods is not the place to disclose results. Furthermore, both metrics are part of results in Tables 10, 11, 12 and Figure 10, therefore, this statement is questionable.
  12. The results in Table 1 and Table 3 are presented as relative errors. It is, however, questionable how to interpret these numbers as they are neither presented in percentages from the reference value nor in the normalized range (0-1). Please explain, or recompute the values in %, everywhere appropriate.  
  13. The caption of Table 2 does not present the interpretation of ‘*’ ‘**’ ‘***’ in a comprehensive manner.
  14. Ln 282: “Figure 4 shows the coverage from AppleWatch dataset.” -> Clarify the statement. Correct the term “coverage” in all instants in the text. The next statement is not ideally correct, because I see differences between curves in Figure 4,
  15. It is not clear how the results in Figure 4 are obtained. Furthermore, here the permitted error on the x-axis is measured in (%)  while Table 1 and Table 3 present the term “Relative error” as a ratio. The question is noted in the comparison of Table 6 and Figure 6, Table 9 and Figure 8. Clarify if figures and tables present the same type of information but on different datasets.
  16. The discussion is detailed for this method, however, I don’t see comparison to the results of other similar methods for R-beat correction. Normally discussion encloses an informative table, where several studies are compared, including definition of their methods, experimental settings and numerical results. This helps in a comprehensive and justified manner to compare the achievements in this study with others in the literature.

Author Response

(The authors gave the same response as above.)

Reviewer 3 Report

The authors present a segment-based gap-filling method to address data loss by low signal quality and motion artifacts for HRV analysis. The authors found best-performing correction depends on the analyzed HRV metrics. I would like to see a revision.

(1) RMSSD and SD1 are not defined in the abstract.

(2) Which HRV metric is the best indicator for accuracy and why?

(3) For alpha = 1.1 (line 175), why and any reference to support?

(4) line 207: "only subjects with respiratory rates above the classic LF band (> 0.15 Hz) have been selected", is this reasonable, and do the authors have any reference to support it? what's the age and gender distribution for the selected subjects?

Author Response

(The authors gave the same response as above.)

Round 2

Reviewer 1 Report

The authors addressed the concerns raised in the previous review. However, some minor change is still needed for clarity.

1. Line 149: please change "the remaining central minute of the segment" to "the remaining segment"

2. Equation 3: please change the "<" to ">". It is better to make the statement positively. i.e. "mark as an outlier if the equation is satisfy". 

3. Line 379: add "," after following

4. The conclusion did not convey the conclusive statements. It just described what had been done. e.g. "Furthermore, the performance analysis allows to extract some conclusions about when to discard a segment for further analysis depending on how much error is assumable in the specific application." That statement is not conclusive. This section can be remove.

Author Response

We thank the Reviewer for the helpful and constructive comments, which have contributed to the improvement of the manuscript. Please see the attachment.
